



# Yardangs and Dunes : Minimum- and Maximum-Dissipation Aeolian Landforms

Ralph D. Lorenz[1]

[1]Johns Hopkins Applied Physics Laboratory, Laurel, MD 20723, USA

*Correspondence to*: Ralph Lorenz (ralph.lorenz@jhuapl.edu)

**Abstract.** Yardangs are ridges formed in soft rock by aeolian erosion in a unidirectional wind environment, and often have a 4:1 length:width ratio that is associated with a minimum-drag shape for a given width. Dunes are emergent aeolian landforms formed by accumulation and removal of sand particles. Dunes have a range of morphologies (barchans, stars, linear, transverse etc.) which can be mapped to the sand supply and the diversity of wind directions. The dune pattern that generally emerges is one that maximizes gross bedform normal transport (GBNT). For fixed imposed wind speed, a minimum drag force corresponds to a minimum dissipation, whereas maximum sand transport corresponds to maximum dissipation. These examples illuminate a more general paradox in non-equilibrium thermodynamics, wherein entropy production rates may be maximized or minimized depending, vaguely, on the degrees of freedom in the system. In these geomorphological examples, however, the difference is informatively clear : whereas yardangs emerge simply by removal of material alone and dissipation is minimized, dunes form by the much less constrained removal and accumulation to maximize dissipation..



## 1 Introduction

Science seeks to explain and predict the sometimes bewildering complexity of nature through unifying principles. In some settings, typically with few interacting entities, the observed behaviors can be captured with a few simple laws, most notably the

motion of planetary bodies around the sun. Yet when the number of elements proliferates, simple deterministic laws become challenged. Where the number of elements is so large as to defy explicit quantification (as in the distribution of molecules of gas in a room), statistical methods must be employed. Thermodynamics, at its root, is merely an expression of statistical probabilities – the molecules will be uniformly distributed, simply because that is the configuration that has the most number of ways of being true, that which maximizes the entropy of the system.

The 2nd law can be generalized to suggest that a system may be most likely to be found in a configuration that maximizes the production of entropy (e.g. Dewar, 2003). This principle of Maximum Entropy Production (MaxEP, or less unambiguously, MEP) was originally proposed for the Earth's climate (e.g. Paltridge, 1975; Ozawa et al., 2003, Kleidon and Lorenz, 2005). The principle has been shown to be usefully applied to Mars and Titan (Lorenz et al., 2001), wherein horizontal heat transports in the atmosphere driven by the temperature contrast that results from uneven insolation reduce that temperature contrast, but only

modestly. The product of heat flux and temperature difference (divided by absolute temperature) corresponds to the maximum (Carnot) thermodynamic work output of the system, or in steady state, to a maximum dissipation. This state is close to (and with present data cannot be discriminated from) the MaxEP state. Other entropy (or force-flux product) maximization applications have been suggested, such as in crystal growth (Martyushev, 2007), fluid pipeflow (Niven, 2010), ecological systems (Kleidon et al., 2008) and earthquake dynamics (Main and Naylor, 2008).

A persistent challenge is the well-established literature (e.g. Prigogine, 1967; Jaynes 1980) noting that many systems are forced to be in states of minimum dissipation or MinEP. Proponents of MaxEP recognize that these are typically systems with few degrees of freedom wherein the state is easy to predict in other ways anyway, whereas MaxEP seems to apply where the system has many degrees of freedom, and many possible steady states, among which MaxEP may be a usefully predictive selection criterion. This distinction between constrained and 'free' systems, however, has not been succinctly, generally and persuasively

defined (although see section 4), and MaxEP cannot be considered useful until its domain of applicability has been determined. This paper offers two geomorphological examples of landforms that correspond to extrema in the dissipation associated with their formation in the hope of elucidating this distinction. Remarkably, these two landforms can locally coexist, yet with completely orthogonal orientation (figure 1).

[Figure 1]

## 45   2 Yardangs

Yardang is a Turkic word that refers to a ridge formed by aeolian (wind) erosion (e.g. Blackwelder, 1934; Goudie, 2007). Although they can occasionally form in hard rock, they typically form in rather friable lake sediments (i.e. consolidated mud and sand) or ignimbrites (volcanic ash deposits). Because sand driven by the wind is the agent of erosion, rather than the wind itself, the sand must be mobile and thus this landform is associated with desert regions. (Not only must there by mobile particles to act

as wind-driven tools of erosion, but wind-blown erosive processes must be more effective than fluvial erosion which would otherwise dominate to form more familiar valley systems.) Among the most prominent lake sediment examples (e.g. figure 2) are in the deserts of western China, North Africa (notably Egypt and Tunisia) and the Lut desert in Iran. Some small examples



(e.g. Ward and Greeley, 1984) are found in the southwestern USA, e.g. near Rogers Lake in California. Ignimbrite examples are found in the Andes, in Peru and Argentina (e.g. Da Silva et al., 2010). Examples are also seen on the surface of Mars (in the

Medusa Fossae formation, assumed to be friable ashfall – Ward, 1979; Zimbelman and Griffin, 2010) and in one possible locale on Venus.

[ Figure 2]

A wide range of scales of yardangs are found, from a couple of meters to many kilometers long. While the length:width:height ratios can vary substantially, most evolved yardangs tend to have a teardrop shape with the ratios 10:2:1– motivating the comparison with upturned boat hulls. Ward and Greeley (1984) noted that the shape was close to that which minimizes fluid dynamic drag, which is after all a consideration in boat design.

[ Figure 3]

Clearly, fresh yardang systems where a wide mass of soft rock has only recently been exposed to aeolian erosion, may take the form of long unbroken ridges. The end state of the system is with the structures entirely eroded away. However, the process is asymptotic, and evolution will slow when in a lower drag state and hence observation at a random moment is more likely to

observe it in such a state. This explains the apparent predominance of 3-5:1 length:width aspect ratios – see figure 3. This figure illustrates the planform evolution of a small soft rock analog exposed to erosion by sand in a wind tunnel in an experiment by Ward and Greeley (1984). The initially broad lump of rock is initially slimmed by erosion, since facets more exposed to the flow are more rapidly eroded (e.g. Bridges et al., 2004). Some irregularities develop, since stochastic fractures may remove more material than is 'optimal' at any moment, but by 100 hours the block has attained a near-perfect streamlined shape, and its

subsequent evolution is very slow, barely changing (but decreasing slightly in size) over the following 100 hours of erosion.

It is evident that the view of yardangs as minimum-drag shapes is somewhat simplistic. First, the 4:1 optimum noted by Ward and Greeley (1984) is a purely fluid concept that may apply over only a restricted Reynolds or Mach number range, that might be exceeded in other environments. Second, it is not a pure fluid that performs the erosion, but rather particles advected in the

stream, so particle diameter (which influences an inertial length scale via the mass:area ratio of the particle) may control how 'streamlined' a yardang of a given size may get. There may be some useful studies to be performed, via observation of extreme conditions such as those at Mars, or by experiment, that expose such dependencies. Finally, as always in geomorphology, the observed landscape reflects a convolution of initial conditions and past and present processes, which can be sometimes difficult to disentangle. Nonetheless, the core idea of Ward and Greeley (1984) that aeolian erosive systems dwell in the 4:1 region of

morphology phase space, which is close to a minimum in dissipation, seems to be usefully accurate.

## 3 Dunes

Dunes are constructive landforms (figure 4) generated by the transport of sand by wind (e.g. Bagnold, 1941; McKee 1979) found widely on Earth and Mars, in a couple of locations on Venus, and in a wide equatorial belt on Titan (e.g. Lorenz and Zimbelman, 2014). While they are fundamentally depositional in character, there is simultaneous removal of material in other parts of the

dune. In some seasonally varying wind regimes this leads to a periodic fluctuation in the dune shape, with a slip face being





developed on alternate sides of linear or reversing dunes. Where the wind direction is more uniform the removal and deposition leads to a net migration of the dune across the landscape, retaining a constant shape as it does so.

[ Figure 4]


While many key aspects of dune mechanics were laid out by Bagnold (1941), a much fuller quantitative understanding has been developed in recent years as a result of careful experiments   (e.g.  Rubin and Hunter, 1987; Rubin and Ikeda, 1990) and numerical simulations (e.g. Reffet et al., 2010).  The mapping of dune morphology to the wind direction diversity (figure 5)  is now very well established, and is generally (with a recently-discovered exception to be discussed later)  follows a Gross Bedform

Normal Transport (GBNF) criterion   (Rubin and Ikeda, 1990).  Thus in unidirectional winds the crestline forms orthogonal to the wind  (except in the case of barchans where there is not enough sand to make a continuous ridge).   In reversing winds, similarly, the dune crestline is orthogonal to the wind direction.

However, in winds which alternate between two obtusely-converging directions, so-called linear dunes form, where the long axis

of the dune aligns roughly along the vector sum of the two directions (i.e. longitudinally along the net transport direction).  Thus while the dune is along the net (long term average) transport direction, it is largely orthogonal to each of the two  instantaneous transport directions, maximimizing the GBNF.

[ Figure 5]


Beyond morphology, this criterion can be seen to be consistent with dune size and number. For a given amount of sand in a small domain subject to unidirectional winds, the maximum dissipation state will be one in which all the sand is piled in a steep ridge orthogonal to the flow.  Clearly, sand cannot be piled steeper than the angle of repose, and the lee side of a dune (the slip face) indeed reaches this angle.  Indeed, generally, dunes grow.  Ultimately their size tends to be limited by the height of the

atmospheric boundary layer (Andreotti et al., 2009), which acts as a capping surface rather like the surface of a liquid, resisting upward displacement and causing higher shear stress as the crest approaches the top.

There is of course a wider variety of dune shapes than the half-dozen major categories described above.  Some such as climbing or echo dunes are associated with other constraints such as topographic obstacles. In other cases, such as the Martian  teardrop

barchans in figure 5a, the form results from a particular combination of wind directions and frequencies, as elucidated by water flume experiments (e.g. Taniguchi et al., 2012).    In some cases, the observed dune pattern is out of equilibrium, in the process of adjusting from one regime to the other.  Furthermore, it often happens that small (more recent) dunes are superposed on large ones, before the total sand volume has had time to respond to a new regime.   This system memory accounts for much of the complexity of observed dune forms, but the underlying GBNF criterion is well-established as the fundamental basis.

**4 Thermodynamic Analogy**

Entropy minimization has been invoked as a concept in thermodynamics, more usually as an academic exercise than as a tool with predictive utility.  For example, a metal bar heated at one end will have a temperature distribution governed by the boundary conditions placed on it, and the cross-section and thermal conductivity of the bar.  In the case where a uniform bar is insulated and held at a constant temperature at each end, the bar will simply have a constant temperature gradient, and if



perturbed from this condition will eventually relax back to it. Such a temperature distribution can be shown to minimize the production of entropy (e.g. Prigogine, 1967).  However, given the boundary conditions, such a steady-state solution follows explicitly, and uniquely, from the conservation equations determining heat flow, so the invocation of extremisation principles adds nothing new.

In contrast, consider the turbulent fluid (atmospheric and oceanic) heat transports in the Earth's climate system.  Low latitudes receive more sunlight than high, and therefore have higher average temperatures,  but since the outgoing infrared flux to space is a function of temperature, and heat can be exchanged between regions (the problem can be visualized simply in a 'two-box' climate model, Lorenz et al., 2001; Lorenz, 2010) there is no unique solution.  In the limit of no heat exchange, there is a large temperature gradient; when turbulent heat transfers are large, they compensate for the insolation imbalance and the planet is
isothermal.  Factors such as planetary rotation or a very thin atmosphere can constrain the solution to have small heat transfer, but in general the system is not determined.  It has been noted empirically that the terrestrial climate  appears to be in a state where the heat transfer is such as to maximize the production of entropy by that heat flow.  In other words, the system 'chooses' (or rather, is most usually found in) a state with intermediate heat flow which is large, but not so large as to destroy the temperature gradient that drives it.  Thus the product of the temperature gradient and the heat flux is maximized : physically this
corresponds to a maximum in entropy production, or near-equivalently to a maximum in mechanical (Carnot) work output. In a steady state, a maximum work output is balanced by a corresponding maximum in dissipation.   This concept has considerable appeal, but has remained controversial.

One source of confusion in assessing the concept is the widespread knowledge of the entropy production minimization
associated with constrained systems as described above, which apparently contradicts the maximization postulated for the climate system.  The distinction likely lies in the extent of the constraints.  Jupp and Cox (2012) have made some progress in identifying how planetary rotation may act as a constraint, controlling planetary heat flows for high values, but allowing a MaxEP state when the atmosphere is thick or slowly-rotating enough.  A particularly extensive discussion of MinEP vs MaxEP and other principles is that by Niven  (2010), who notes (crudely) that a fixed-flow pipe system requires MinEP, but a fixed-head
system chooses MaxEP.  We have shown in this paper that an analogous dichotomy exists in aeolian landscape evolution, where yardangs apparently minimize frictional dissipation, whereas dunes maximize it.  We note that minimum-dissipation concepts have long been applied to fluvial landscape evolution (e.g. Sun et al., 1993) where presumably fixed precipitation fluxes operate analogously to wind and sand on yardangs networks found in nature.

A recent elaboration in dune morphology has been discovered, which identifies a 'fingering' mode of dune growth (Courrech du Pont et al., 214; Gao et al., 2015). This occurs in cases where the sand supply is limited, and typically results in a downwind extension of the bedform – essentially forming a 'tail'. Similar morphologies are seen in cases where the sand is 'sticky' – e.g. Rubin and Hesp (2009).  These cases represent instances where the 'traditional' dune quasi-equilibrium of deposition and removal breaks down, and the landform results essentially only from deposition. This scenario then embodies again (but at the
other end of the spectrum – figure 6)  a constrained situation where the minimum-dissipation morphology emerges.

[ Figure 6 ]



## 5 Conclusions

While the present paper offers no formal definition in the degree of constraint of the two landforms described, it has noted the apparently opposing extremisation tendencies and suggested that the freedom afforded by combined construction and destruction allows dunes to maximize transport with edges normal to the wind, whereas yardangs erode into streamlined minimum-drag forms.  Half a century ago, entropy has been invoked as a concept in landscape evolution (Leopold and Langbein, 1962) : it is suggested in this paper that further useful applications in this domain may be found.

## 175    6 Acknowledgements

This work was supported in part by NASA grant 80NSSC18K1389, "Cassini Radar Science Support".

### Code/Data availability

No code or data were used in this concept paper


### Author contribution

RL developed the concept and wrote the manuscript

### Competing interests

None

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




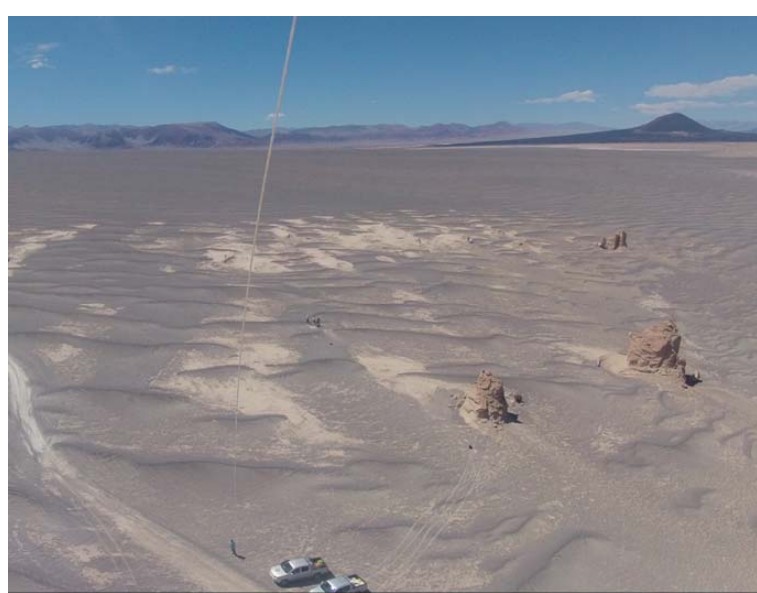


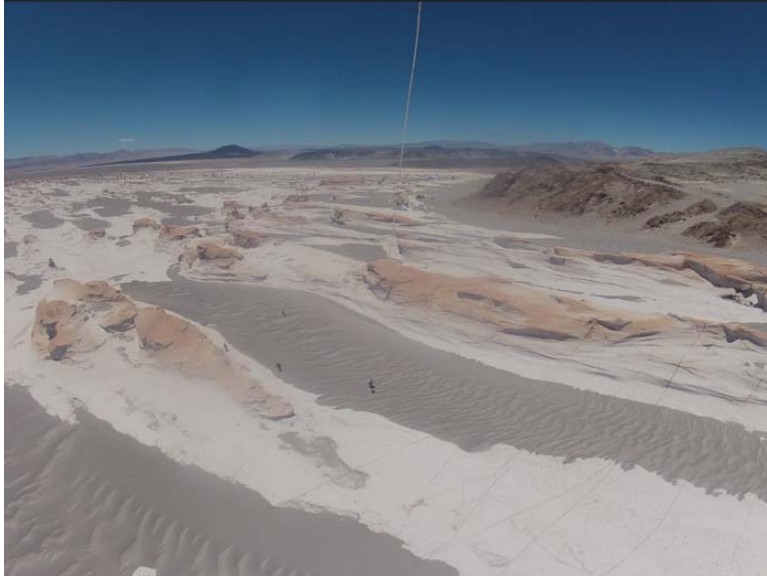

**Figure 1: Two kiteborne views of yardangs and dune-like gravel ripples in the Argentine Puna at an elevation of 3000m. The yardangs erode into streamlined forms aligned with the main wind direction, while the ripples align at right angles to it. This**
**juxtaposition of orthogonality highlights the paradox addressed in this paper (Photos : R. D. Lorenz)).**



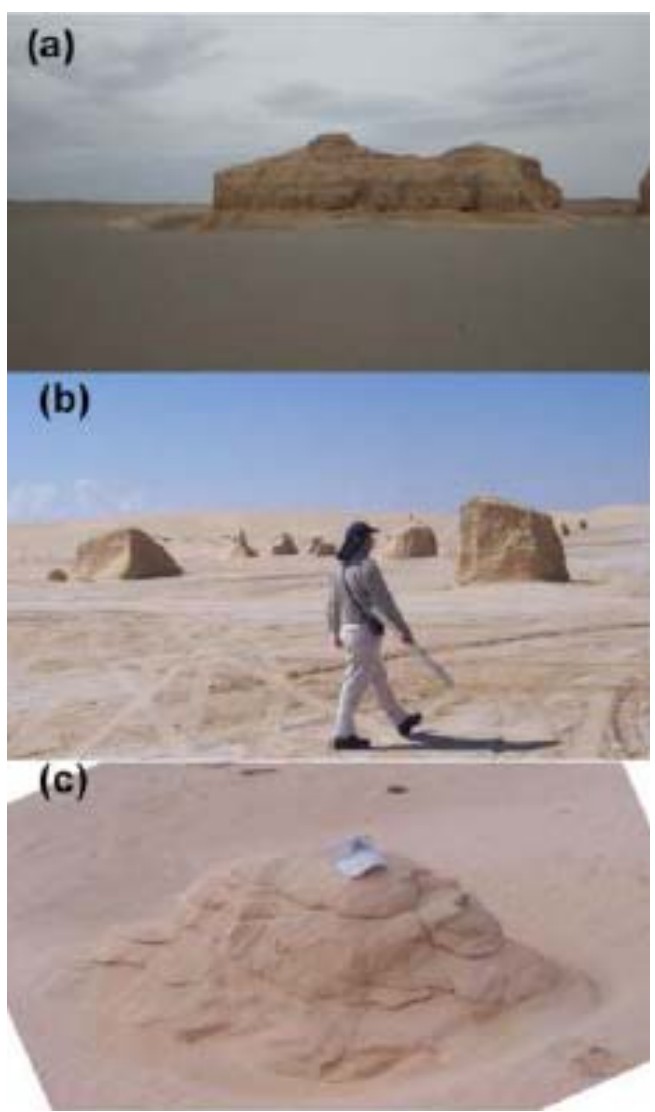

**Figure 2: Yardangs, showing the typical 4:1 aspect ratio. (a) a large example in the Yardang Geopark near Dunhuang, Gansu**
**Province, western China. This example is about 10m tall – note the scoured depression at the nose of the yardang at left (Photo: R.**
**Lorenz). (b) Small yardangs on the Chott El Gharsa lakebed in Tunisia : yardangs may be seen in the movie "Star Wars: Episode 1",**
**Lorenz et al., 2013 (Photo J. Radebaugh). (c) A small remnant yardang, showing the classic streamlined shape – hat for scale**
**(Photo: J. Radebaugh).**




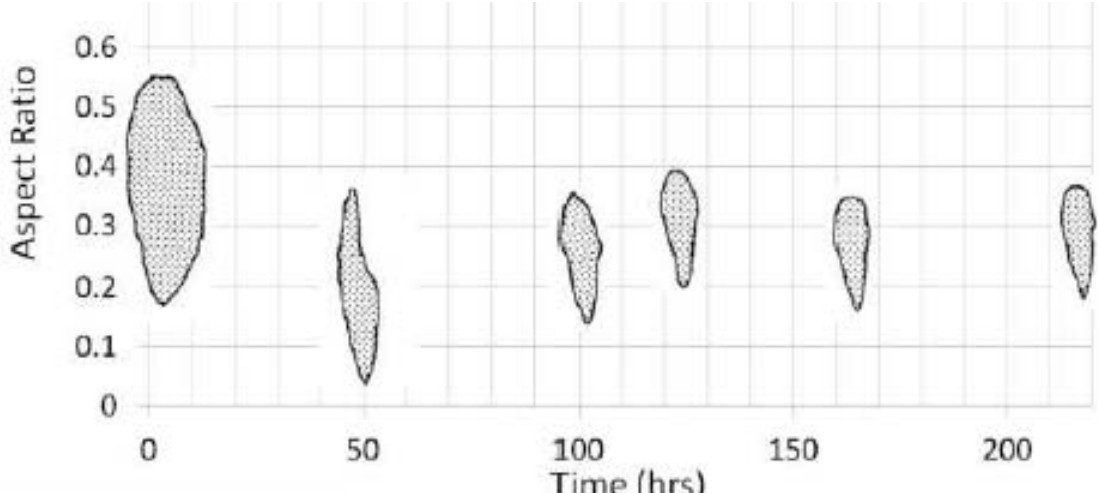

**Figure 3.** **Evolution of the shape of a soft-rock block exposed to a sand stream in a wind tunnel by Ward and Greeley (1984). The block initially slims and shrinks rapidly, but soon attains a streamlined shape which then evolves very slowly. Observing this block at a random instant, one would most likely encounter the streamlined teardrop shape.**



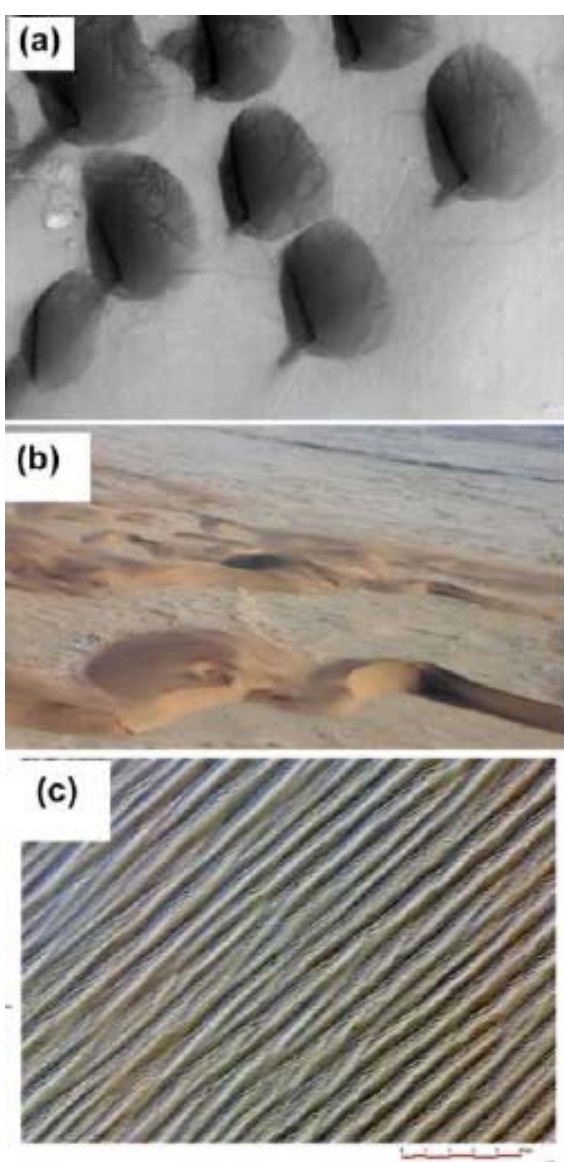

**Figure 4. Some examples of the diversity of duneforms encountered on planetary surfaces. (a) spacecraft view teardrop barchans on**
**Mars, a morphology rarely found on Earth. Note the dark tracks made by dust devils on the dune surfaces (photo: NASA/U.Arizona)**
**(b) More conventional hooked barchans, in the Namib sand sea near the Atlantic coast (photo: R. Lorenz) (c) satellite view of linear**
**dunes in the Rub' Al-Khali desert in Saudi Arabia : smaller linear dunes are superposed on the main pattern (photo: ASTER image**
**by NASA/JAXA/METI).**






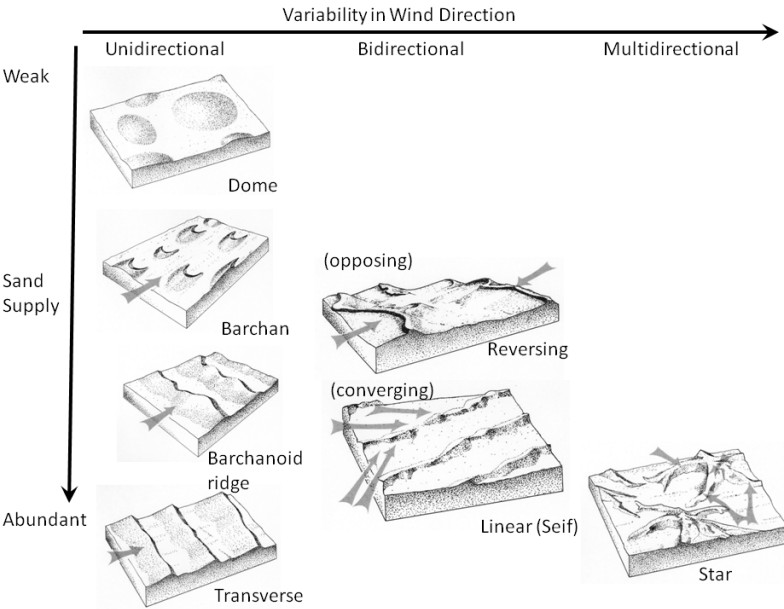

**Figure 5. Major dune types, as defined by McKee (1979) mapped out on a diagram of sand supply and wind variability. Note that the dune crests are aligned in way that maximizes the sand flux normal to the crest, a principle of particular interest in linear dunes where two obtusely-converging wind directions apply.**




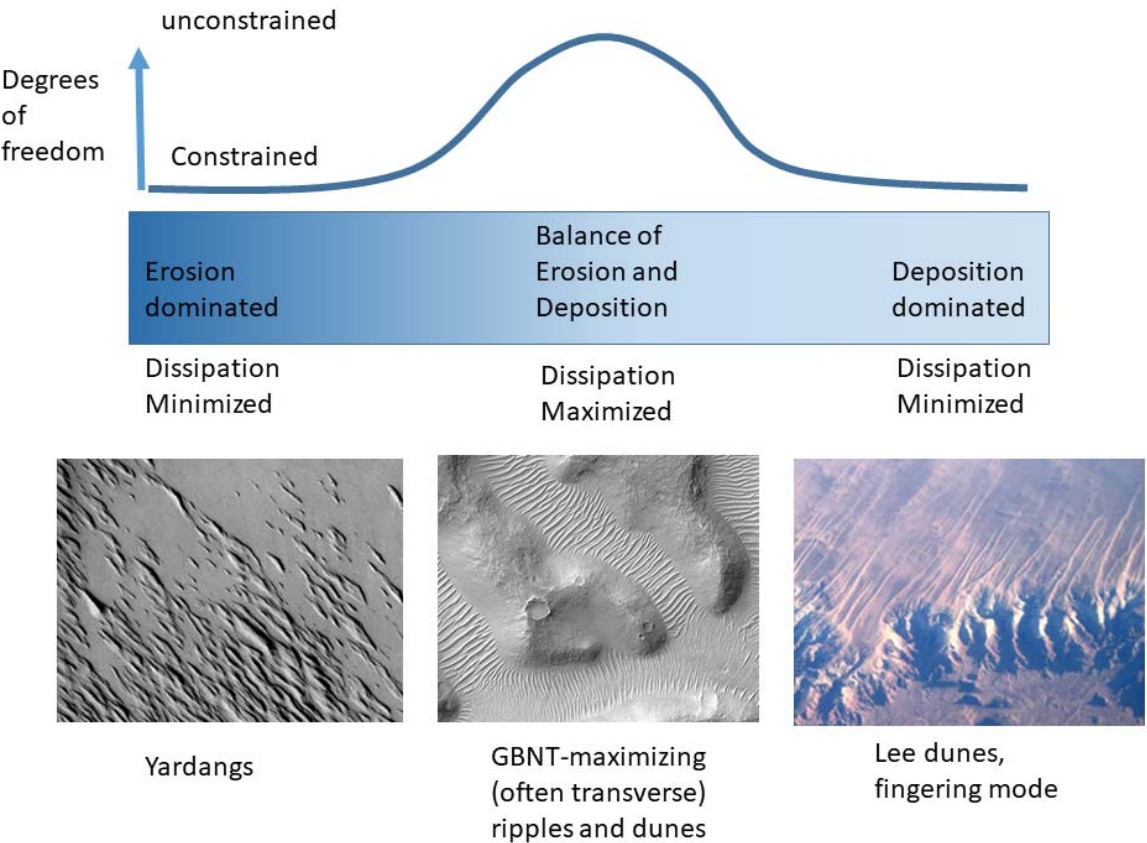

**Figure 6.** **Schematic of the spectrum of the erosion/deposition balance, the associated 'degrees of freedom' and the resultant dissipation-extremization and landforms.**