# Peer review of "Yardangs and Dunes : Minimum- and Maximum-Dissipation Aeolian Landforms"

_Earth System Dynamics, 2019_

## Short Comment (SC1) · 5 Dec 2019

I noticed that this manuscript does not fulfil the criteria of an ESD Ideas manuscript, as it is much too long. ESD Ideas manuscripts need to be short (about 2 printed pages, 1 figure), as described on the web page (https://www.earth-system-dynamics.net/about/manuscript_types.html).

After consultation with Copernicus, we decided to switch the submission category of this manuscript to a regular research article. This means in the review process, it should also be treated as a regular submission, with the associated review criteria.

I apologize for not catching this before this submission entered the discussion.

Axel Kleidon

---

## Referee Comment (RC1) · Hisashi Ozawa (Referee) · 21 Jan 2020

General comments

This paper investigates the formation process of two typical landforms (yardangs and dunes) and their thermodynamic properties associated with the landform formation. It is shown that yardangs, which are formed by a wind-driven erosion process, tend to take the form that minimizes the wind drag force and minimizes kinetic energy dissipation due to the wind drag force. In contrast, dunes, which are formed under the balance between the erosion and accumulation processes, tend to take the form that maximizes the wind drag force and maximizes dissipation of kinetic energy of turbulent flows associated with the dunes. While the physical mechanism that determines

the significant difference between the two distinctive landforms is not fully explained, this paper provides a good introductory review on this subject and can be regarded as a thought-provoking, informative paper. The text is very well written, and there is no serious technical error in this paper. I therefore recommend publication of this paper. However, I have several comments and suggestions that may be helpful for improving the quality of this paper before the publication.

Specific comments

1. Line 46: Yardangs is a Turkic word.

I am not sure, but someone says it is Uighur (e.g. Wang et al. 2011, PRE). It may be good to check the word's origin.

2. Line 114-115: their size tends to be limited by the height of the atmospheric boundary layer.

The atmospheric boundary layer is defined as the lowermost atmospheric layer that is influenced by the surface drag force. So, the height can vary with the surface topography - as the density and height of dunes increase, the BL height may become higher. If so, I am not clearly sure if they can really be "capped" or limited by the variable BL. Can you make this point clearer?

3. Line 132: heat flow.

Here it may be good to state that this heat flow is conduction only. In this case, the governing equation is of the Laplace type (without a nonlinear advection/convection term), and everything is solvable.

4. Line 138: there is no unique solution.

There is no unique solution because the heat transport rate for a movable fluid includes a nonlinear advection term - the heat transport rate can change according to a change in dynamic motion, whose analytical solution has not yet been obtained.

5. Line 154-155: a fixed-flow pipe system requires MinEP, but ... chooses MaxEP.

Probably, the situation is opposite: a fixed-head system prefers MinEP whereas a fixed-flow system exhibits MaxEP - please check Robert's paper. I do not know the reason, but please note that the transition in this case refers to a laminar to turbulence transition, not a selection among multiple steady sates of a turbulent fluid system.

6. Line 165 (Fig. 6): a constrained situation where the minimum... emerges.

Perhaps, this is one of the most important findings of this paper. This idea is interesting. But, I have a slightly different opinion on the yardang/dune/finger formation.

Yardangs tend to take a form that reduces the wind drag force because they are produced by a one-way erosion process from the wind (+ sands) to the solid surface. The erosion rate, which is roughly proportional to the wind drag force, is largest at edge/vertex parts of an obstacle exposed to wind. The edge/vertex parts will thus be eroded away and tend to be 'flatten' as time goes on - so we observe a streamlined shape for yardangs in due course. The erosion process is one-way (nearly linear), and there is no (nonlinear) feedback from the eroded sand particles. Note also that yardongs are NOT in a steady balanced state, but in a transient eroding state: they will eventually disappear when we observe them over a very long time period (t -> infinity). On the other hand, dunes tend to take a form that increases the wind drag force because they are produced through mutual interactions between wind fields and sand-accumulated surfaces. Their shape is determined not only by a simple erosion process but by a balance between the erosion and accumulation processes - the wind field determines the erosion rate, but the eroded sands accumulate on a land surface which largely affects the wind velocity and the drag force by producing a certain dune shape. There is a kind of mutual interaction between wind and sand surfaces with non-linear feedbacks associated with large and distinctive turbulent eddies adjacent to the landforms. In this situation, the system tends to be in a state with MaxEP as we have seen in many other similar examples (thermal convection, shear turbulence, general
circulations, etc.).

From this viewpoint 'finger' dunes can be seen to be a special limiting case of an initial development under limited sand supply. There seems to be only a one-way process from wind to the sand surface (finger), and there may be no sands-to-wind feedback during a developing stage of finger dunes. In a matured stage, however, a finger dune tends to disintegrate into a series of smaller dunes under unidirectional wind conditions (Gao et al. 2015).

7. Line 234- (references).

Some literature (Taniguchi et al., Ward and Greeley, Niven...) seems to be missing from the list. Please check.

---

## Author Comment (AC1) · 24 Jan 2020

I am grateful for Dr. Ozawa's comments, and will attend to the various minor points raised.

On the matter of figure 6 and the overall framework, I indeed recognized that yardangs are ephemeral in that since material can only be removed, once yardangs adopt the traditional shape, they only shrink and disappear. Thus the right-hand side of the supply/removal balance is not a true equilibrium, but only a quasi-steady state that asymptotes to oblivion.

The left-hand side, with 'supply-only' accretion of material in the 'fingering' mode, is similarly not stationary, as the sand inventory continues to increase.

[Figure]

Only the intermediate scenario, with supply and removal in balance, is a dynamic equilibrium that can remain constant in time.

So I wonder then if this paradigm can offer some guidance on the behavior of fluid systems such as ocean circulation, atmospheric dynamics etc. with non-steady forcing. E.g. when heating is suddenly applied to a fluid system (and its ability to reject that heat has not yet developed) does it grow in a minimum-dissipation configuration until a maximum-dissipation steady state is reached? The discussion paper esd-2019-52 by Kleidon et al is relevant in this respect.

---

## Referee Comment (RC2) · Anonymous Referee #2 · 23 Feb 2020

General comments:

This manuscript discusses the role of thermodynamics in shaping yardang and dune landforms. Yardang is formed by erosion that minimizes the dissipation whereas dunes are formed by the accumulation of sand that maximizes the dissipation. Both of them are contrasting landforms that are mainly shaped by wind. The author describes their individual characteristics very well with captivating pictures, although it is difficult to demarcate the author's edition to the figures that are adapted from literature. Understanding these landforms with examples from other aspects of thermodynamics especially in atmospheric and oceanic science is effectively done. Reading the manuscript feels like reading a book chapter on 'thermodynamic of landforms'. However, the manuscript needs to be clearer on the novelty of the research. Also, the manuscript targets read-

ers from the multidisciplinary background so the author must simplify his explanation and elaborate more on the connection and analogy provided in the introduction. The reader finds it difficult to get the actual message and new findings but the basics and connection to old literature are well presented. This manuscript serves, as a good review on the role of thermodynamics for yardangs and dunes but additional information is needed to transform it into an informative manuscript. I suggest a major revision in the manuscript especially highlighting the main message.

Specific comments:

1) Title: If the aim of the manuscript is to present contrasting thermodynamic of two landforms one can replace "and" with "versus" such that the title is "Yardangs versus Dunes: Minimum- and Maximum-Dissipation Aeolian Landforms". If not then it needs to be clarified why these two landforms are chosen, what is common and distinctive between them in the introduction.

2) Abstract: The abstract needs to discuss less of the literature but more of the author's findings and interpretation. Currently, the first 6 lines are defining the two landforms and other 4 lines present the ideas of the author, but nothing concrete comes out from it. It might be helpful to write " Here I show... already in 3rd line".

3) Introduction: The introduction is attractive however it lacks to connect the author's work to the given examples. Only line 41-43 speaks author's aim. The second paragraph (lines 25 to 34) can be condensed to two sentences and the author can already introduce landforms in the second paragraph by associating it with examples.

4) Introduction: Line (37 to 39)" MaxEP seems to apply where the system has many degrees of freedom, and many possible steady states, among which MaxEP may be a usefully predictive selection Criterion", It will be helpful if you can explain what are the degree of freedoms and steady states for yardangs and dunes?

5) Introduction and figure: (line 42) "these two landforms can locally coexist". Can you

elaborate on why it is remarkable? Also please editing, what feature is yardang, what is a dune and how they are oriented orthogonally in figure 1. I suggest inside the image mention these features and draw their orientation within the figure. This can make the images scientifically important.

6) Line (50 to 51)"(Not only must thereby mobile particles to acts wind-driven tools of erosion, but wind-blown erosive processes must be more effective than fluvial erosion which would otherwise dominate to form more familiar valley systems.)" I wonder why it is inside bracket. Can you transform it into a sentence?

7) Line 51, you refer to figure 2, It will be good to illustrate the aspect ratio in the figure also for each picture (a, b, c). The figure caption (line 251) "Star Wars: Episode 1", please provide reference and time of the clip. Explain the relevance to it in the text.

8) I particularly like the inclusion of other planets (Mars and Venus) in the text. Additional attention is needed in the transition from line 55 to 60. For example Line 60, "A wide range of scales of yardangs are found, from a couple of meters to many kilometers long" I guess this statement was for Earth.

9) The methodology part of the paper is missing. I suggest to write it briefly after the introduction where you explain, where you take pictures, what is the aspect ratio, GBNF and mainly what is your approach

10) Figure 3, you write in the text an aspect ratio of 4:1 but in the figure, it appears in decimals. Consistency is required.

11) Line 113, this can be elaborated better with a figure, maybe one of the images you clicked?

12) The paragraph, especially lines 135 to 147, is great to bring the attention to maximum entropy production but requires a connection to yardang and dunes. Maybe you can discuss how one can interpret it for landforms.

13) Lines 156 to 158 are useful; I would like this kind of information related to fluvial

landforms to be reported more often as it really connects to your approach.

14) I like how you end the conclusion (Line 172 to 173). Can you add future outlooks to it to make it more effective?

15) Figure 6, what is on the y-axis? Is it theoretical or based on some numbers from the literature? How did you draw the Gaussian curve?

Technical comments:

Overall the writing is attractive and has no technical errors. But figures need additional technical information or description in the text (as mentioned above). In some figures, the subfigures are illustrated by a, b and c but in some figures, they are not.

---

## Author Comment (AC2) · 30 Mar 2020

It is gratifying that both reviewers Ozawa and #2 consider the paper well-written and a good review of the topic. Neither of the reviewers challenges the thesis of the paper, that an analogy exists between the landforms formation in balanced (constructive/destructive - dunes) or unbalanced (constructive-only, fingering dunes or destructive-only, yardangs) settings and the emergence of maximum vs minimum dissipation (EP) states in thermodynamic systems.

Reviewer 2 calls for some more-or-less editorial changes, to sharpen the text and to elaborate some explanations and the specific contribution of this paper, as well as some minor figure fixes.

[Figure]

Reviewer 2's suggestion to revise the title, to use 'versus' to underscore the contrast is a good one.

Reviewer Ozawa points out some aspects which similarly need clarification, as ell as a couple of missing references.